# Detection of Physical Activity Using Machine Learning Methods Based on Continuous Blood Glucose Monitoring and Heart Rate Signals

**DOI:** 10.3390/s22218568

**Published:** 2022-11-07

**Authors:** Lehel Dénes-Fazakas, Máté Siket, László Szilágyi, Levente Kovács, György Eigner

**Affiliations:** 1Physiological Controls Research Center, Óbuda University, Bécsi út 96/b, H-1034 Budapest, Hungary; 2Applied Informatics and Applied Mathematics Doctoral School, Óbuda University, Bécsi út 96/b, H-1034 Budapest, Hungary; 3Institute for Computer Science and Control, Eötvös Lóránd Research Network, H-1111 Budapest, Hungary; 4Biomatics and Applied Artificial Intelligence Institution, John von Neumann Faculty of Informatics, Óbuda University, Bécsi út 96/b, H-1034 Budapest, Hungary; 5Computational Intelligence Research Group, Sapientia Hungarian University of Transylvania, 540485 Tîrgu Mureş, Romania

**Keywords:** diabetes mellitus, machine learning, physical activity detection

## Abstract

Non-coordinated physical activity may lead to hypoglycemia, which is a dangerous condition for diabetic people. Decision support systems related to type 1 diabetes mellitus (T1DM) still lack the capability of automated therapy modification by recognizing and categorizing the physical activity. Further, this desired adaptive therapy should be achieved without increasing the administrative load, which is already high for the diabetic community. These requirements can be satisfied by using artificial intelligence-based solutions, signals collected by wearable devices, and relying on the already available data sources, such as continuous glucose monitoring systems. In this work, we focus on the detection of physical activity by using a continuous glucose monitoring system and a wearable sensor providing the heart rate—the latter is accessible even in the cheapest wearables. Our results show that the detection of physical activity is possible based on these data sources, even if only low-complexity artificial intelligence models are deployed. In general, our models achieved approximately 90% accuracy in the detection of physical activity.

## 1. Introduction

Diabetes mellitus (DM) is a chronic disease of the metabolic system related to the insulin hormone. The complete absence of insulin (type 1 DM—T1DM) occurs when the insulin-producing β-cells of the islets of Langerhans in the pancreas are demolished within an autoimmune chain reaction. The insulin is the key hormone of the metabolism, regulating many molecular pathways from which the most important one is the control of insulin-dependent glucose transporters (GLUT), mostly the GLUT-4, which can be found in large quantities in the membrane of adipose tissue and skeletal muscles cells [1]. T1DM occurs mostly in young ages and as a rapid progression of an autoimmune process. In contrast, type 2 DM (T2DM) appears over years as a result of the lifestyle and genetic willingness. Frequently, T2DM remains non-diagnosed for many years and the patients are diagnosed due to a side effect of the malady [2].

A molecular mechanism of insulin through the insulin-dependent GLUT gates is to reduce the blood glucose (BG) level by opening these gates in the cell membrane (usually, these cells where insulin-sensitive gates can be found in the cell membrane are called “insulin sensitive cells”, such as the adipose, skeletal muscle, and liver cells) [2].

Physical activity is a key component of the diabetes therapy (both with T1DM and T2DM). In the case of T1DM, doing exercise on a daily regimen completes the external insulin administration and diabetic diet [3]. Physical activity, more precisely the muscular stress induced by physical activity, is important from multiple aspects: it opens specific non-insulin-dependent pathways in the cell wall through which glucose is able to enter into the cells [4]. Furthermore, it increases the GLUT-4 expression in the skeletal muscle, increases the insulin sensitivity, and contributes to an improved insulin action, glucose disposal, and enhanced muscle glycogen storage, even at 48 h after the exercise [5,6]. The scheduled physical activity—besides its many other positive effects—can help to regulate the blood glucose level and to increase the activity and effectiveness of the metabolic system, making it an ideal and organic part of the diabetic treatment [7]. According to the American Diabetes Association (ADA), at least 150 min per week at moderate-to-vigorous intensity of weekly exercise is recommended for T1DM patients [8]. In addition, resistance exercise is also recommended two to three times a week [8].

Nevertheless, non-scheduled exercise can be dangerous if it is neglected during the therapy. Namely, if the patient does not take into consideration the exercise events when calculating the necessary insulin amounts or fails to introduce the exercise into the insulin pump during pump therapy, it could lead to insulin overdosing, causing serious hypoglycemic periods [9]. Hypoglycemia is an extremely dangerous condition for diabetics. Hypoglycemia with decreasing glucose trends may cause cetoacidic conditions and lead to even coma or death in the short term. Thus, there is an unquestionable necessity to take the physical activity into account in the therapy during daily life and especially if the treatment is semi-automatic, for example, when an insulin pump is applied [10].

In the case of automatic blood glucose regulation, the regulatory algorithms shall take the physical activity into account, and subroutines with the capability of recognizing the exercise events are necessary to avoid a hypoglycemic period, regardless of whether the user reports the physical activity or not. The drop in the BG levels caused by exercise happens with a slight delay, but the effect of physical activity on the BG level regulation stands even 48 h after the exercise event (depending on the type and duration of the exercise) [6,11]. Thus, the dynamics of the exercise effect allow us to intervene promptly in the BG regulatory processes to improve the glycemic state if the effect of the physical activity is recognized.

One of the biggest challenges of the research community is to create such subroutines that can be used to recognize non-scheduled physical activities to support decision making and therapy optimalization for semi-automatic BG regulation solutions. Usually, these systems operate according to the artificial pancreas (AP) concept. The AP systems consist of three components: a continuous glucose monitoring system (CGM) to measure the BG level, an insulin pump as an actuator to inject the insulin, and advanced control algorithms. Generally, the AP systems contain these subcomponents [12,13]. Without additional sensors (e.g., body-worn activity trackers or integrated accelerometer (IMU)/heart rate (HR) sensors in the CGM/insulin pump), the only option to detect physical activity in the case of the users of such systems is to use the CGM signal. The main issue is the delay between the effect of exercise that appears in the CGM signal, making this approach challenging. The IMU and HR signals can be good complements to the CGM signals, because these can indicate the exercise with good accuracy [14,15].

Eventually, physical activity detection, recognition, and, later on, classification based on the type and intensity are key components in T1DM treatment. Many existing solutions are available on this topic in general by using IMU sensors [16]. Recently, the concept of using an IMU to detect and classify physical activity specifically for diabetic patients appeared [17]. The usage of an IMU is also beneficial due to the cardiac autonomic neuropathy (CAN) (with autonomic nervous system (ANS) dysfunction and elevated resting HR) [18]. The CAN—a common long-term side effect in people with diabetes [19]—can weaken the predictive capability of the HR signal in the case of T1DM patients. However, in the short and medium term, the correlation between the CAN, BG level, BG variability (BGV), and, moreover, the HR variability (HRV) is not fully clarified. Furthermore, there are studies showing that the correlation between the CAN, HR, and HRV should be re-investigated [20].

Most of the wearable activity trackers on the market do not provide any access to the raw IMU data [21]. There are devices that provide such access (e.g., Empatica E4 [22]); however, their prices are higher (approx. 1000 USD), making such devices barely available to the diabetic population. Wearable sensors providing HR data on at least a 5 min sampling time basis are quite cheap with easy access to the data directly from the device and, for example, from activity tracking applications [21]. The 5 min basis allows the use of the sampled HR data beside the CGM signal as an indicator of physical activity.

The capability of artificial intelligence (AI) tools to recognize patterns has been proven in many applications related to biomedical engineering [23,24,25,26]. In the case of diabetes treatment, they have also proven their usefulness [27,28,29,30].

In accordance with the previously introduced facts, investigations, and literature, we can conclude that the CGM signal and the complex of the CGM and HR signals are potential targets from which the physical activity can be detected. In this study, our goals are to develop AI applications which are capable of detecting the physical activity using only the CGM signal or the CGM and HR complex on a binary basis (i.e., whether the physical activity happened, is happening, or not). These algorithms can be extremely useful in closed-loop insulin delivery systems. It has to be noticed that we do not categorize the type of the activity, only the recognition of its presence is the goal as a first step in our research.

The remaining part of the paper is organized as follows. Section 2 shows the applied methodology, including the clinical data extraction, classification methods, and metrics used for the performance evaluation. Section 3 exhibits the results obtained by the employed methods. Section 4 discusses the achievements of the capabilities of the involved classification models in various circumstances. Section 5 concludes the study.

## 2. Materials and Methods

### 2.1. Preliminary Results

In a previous study [31], we focused on establishing simple machine learning-based algorithms using synthetic data coming from virtual patient environment to develop physical activity detectors. We applied only the simulated CGM signal and features extracted from the CGM. The tested features were the same as we introduce here in this study. We found various machine learning algorithms that performed well (e.g., KNN, Random Forest, Decision Tree) in the physical activity detection. In this study, one of our goals is to validate the previous conclusions and findings of [31] using real patient data.

### 2.2. Development Environments

In this study, we use Python 3.8 language in a hosted cloud environment. We utilize the following platforms and libraries: Tensorflow 2.0.0 [32], Scikit-learn [33], Numpy [34], Pandas [35]. The implementation is performed by using Jupyter Notebook [36] development user interface. The already mentioned hosted cloud environment is Google’s CoLaboratory^TM^ (shortly “Colab”) [37] where we apply the host provided by Google, which is free of charge by default. The dedicated resource on the platform varies from use to use, albeit it is around 12.69 GB VRAM and 107.79 GB VSPACE with 4 VCPUs provided by a Python 3 Google Compute Engine server.

### 2.3. Datasets

#### 2.3.1. The Ohio T1DM Dataset

The Ohio T1DM dataset is a semi-public dataset available for scientists working in research on T1DM. Originally, the dataset was published in 2018 by Marling et al. [38] for the blood glucose level prediction (BGLP) challenge [39]. In 2020, the dataset was extended by data from further six subjects [40,41].

The dataset includes 8 weeks of data collected from 12 T1DM subjects. Each subject used continuous glucose monitoring system (CGM) and was on insulin pump therapy. The dataset from each patient contains the following information: CGM readings of blood glucose level every 5 min; blood glucose levels from periodic self-monitoring of blood glucose (finger sticks); insulin doses (bolus and basal); self-reported meal times with carbohydrate estimates; self-reported times of exercise, sleep, work, stress, and illness; physiological data from fitness bands and environmental data (e.g., temperature) [41].

In this work we extracted the following data types from the original dataset (complex XML structure):Glucose level: contains the time stamps (date and time) and the CGM data (in mg/dL) recorded every 5 min;Exercise: contains time stamps of the beginning of the exercise (date and time), intensity values (subjective intensities on a scale of 1 to 10, with 10 the most physically active), type (exercise type), duration in minutes, and a “competitive” field;Basis_heart_rate: contains time stamps and the HR data (in bpm) recorded every 5 min;Body weight of the patient expressed in kg (measured once at the beginning of the 8-week experiment).

The greatest challenge in the usage of the Ohio T1DM dataset stems from the fact that all physical exercise is self-reported. We have deeply analyzed the extracted data fields—the results of the examination are introduced in Figure 1. Here, we created a graphical representation of what we have found in the given patient’s recordings. We identified different “islands” in the data structure: mostly, the data and time stamps were available during the whole duration of the experiment; the CGM data were available in “blocks” (with given starting time and duration, usually lasting for multiple days); the exercise events were reported occasionally (with starting time and duration).

We identified some exercise events where there were no corresponding CGM data (e.g., Figure 1—transparent area covered by orange dashed line). We neglected these data (did not extract) because our use-cases require at least the presence of the CGM signal. Those 24 h intervals were also neglected where no physical activity was reported. This was necessary because our investigations showed that, in many cases, the self-reporting of the patients was not accurate and they possibly made physical activity without reporting it (e.g., Figure 1—transparent area covered by red dashed line). Due to this latter issue, the amount of data to be extracted became much smaller. This fact will be significant later on, because the algorithms developed in this study could be used for data annotation, as it is presented in Section 3 and Section 4. To be sure that we extract only those data where the self-reported exercise was unequivocally identifiable from the data, we decided to extract only those smaller 24 h regions (midnight to midnight) where the CGM signal and the reported exercise were both available (e.g., Figure 1—blue and green regions). These data were extracted and used when the modeling goal was to detect physical activity only from CGM signal. Another use-case was to detect physical activity by using the CGM signal and the corresponding exercise events where the HR data with the same date and time stamps were also available (e.g., Figure 1—green region). This kind of data extraction further reduced the size of the usable dataset.

Thus, we have extracted two “cleaned” datasets:(i)CGM data in 24 h long blocks where self-reported exercise happened within the 24 h (e.g., Figure 1—blue and green regions). The exercise event was set to 1 where the exercise was ongoing. We made an exception—where the reported activity level was lower than 2, we set the exercise event to 0. The numerical properties of the extracted dataset can be seen in Table 1.(ii)CGM data in 24 h long blocks where self-reported exercise happened within the 24 h and HR data were also available (e.g., Figure 1—green region). The exercise event was set to 1 where the exercise was ongoing. We made an exception—where the reported activity level was lower than 2, the exercise event was set to 0. The numerical properties of the extracted dataset can be seen in Table 2.

A data record in the case of each patient was in the following structure: [Date stamp, Time stamp, BG level from CGM (concentration), HR value (integer), Exercise (0/1)]. Data records were organized in chronological order. The weight of the patient, because it was reported only once at the beginning of the experiment, was handled separately.

#### 2.3.2. The D1namo Dataset

The D1namo dataset includes measurements of 20 healthy and 9 T1DM subjects collected using wearable devices (Zephyr BioHarness 3) within non-clinical conditions. Only the data of the 9 T1DM patients were involved in this study.

The dataset consists of electrocardiogram (ECG), breathing, and accelerometer data. Moreover, it includes glucose measurements and annotated food pictures as well [42]. Aggregated values are also provided based on the ECG signals, from which the heart rate and the activity level are utilized in our study. The patient’s body weight is also provided.

Generally, the dataset covers a few days of the subjects only (1–3 days overlapping). The sampling time of the CGM is 5 min; however, the sampling frequency of the HR and activity is 1 Hz. Moreover, the data quality is not as good as in the other investigated dataset. There are many inconsistencies and missing values in the CGM, HR, and activity as well. The basis of the data cleaning was that we considered the BG registrations in the following format: [date stamp in YYYY-MM-DD, time stamp in HH:MM:SS, BG concentration in mmol/L, type of measurement (manual/CGM)]. From the HR and activity data, we extracted only those that could be collated to the BG registrations according to the date and time stamps. In this way, for each subject, we obtained CGM data at a sampling time of 5 min and the HR and activity levels data at a sampling time of 1 s. The ECG sensor’s sampling frequency is 250 Hz, and the accelerometer operates at 100 Hz—although, the data stream is given on a 1 s basis. Thus, ∼300 data points for both of the HR and activity level values were given between two BG registration in the extracted data (we experienced some data inaccuracies in some cases). To solve this issue, we simply aggregated the HR and activity levels between the BG registrations (coming from the CGM) using the numerical average of the values ($\frac{1}{N}\sum_{i=1}^{N}{\mathrm{HR}}_{i}$, where N=1∼300 depends on whether there were missing data or not). The activity level is automatically determined by the Zephyr sensor from IMU and HR data. According to [43], if the activity level is higher than 0.2, we can consider a mild activity (with respect to USARIEM guidelines). We considered that the exercise event corresponding to a given record was 1 if the activity level was higher than 0.2; otherwise, we set it to 0.

After the data cleaning, structured datasets can be achieved. Similar to the extracted data of the Ohio T1DM dataset, the final structure of data records in the case of each patient followed this structure: [Date stamp, Time stamp, BG level from CGM (concentration), HR value (integer), Exercise (0/1)]. The data records were organized in a chronological order, the first record belonged to the first recorded data. The weight data were handled separately. Table 3 shows the number of extracted records from the original D1namo dataset using only the CGM data and, separately, the record count with both the CGM and HR features.

### 2.4. Investigated Machine Learning Methods

In this section, we summarize the machine learning methods involved in this study, including their technological foundations. The methods are predefined in the deployed Scikit-learn [44] library where their parametrization can be performed easily. We made different tests during our previous study in this domain [31] where we applied grid search-based methods to determine the most suitable algorithm and its optimal settings. We set the technical parameters of the different methods alongside our previous investigations where we applied synthetic patient data coming from in silico experiments. Because one of our aims with this research was to investigate the classification performance of models created on synthetic data using real patient data, we used the settings previously established for all models and methods in the experiments.
*Logistic Regression* (LR). The LR models provide the probability whether a given sample belongs to a particular class or not by using the logistic function: $f\left(x\right)=M\cdot \exp[k\left(x-x_{0}\right)]$, where *k* is the steepness of the logistic curve, *M* is the maximum value of the curve, and $x_{0}$ is the inflection point [45]. During the training session, we allowed maximum 1000 iterations, and we applied $L_{2}$-type penalty to measure the performance.*AdaBoost Classifier* (AdaBoost) represents a boosting technique used in machine learning as an ensemble method. The weights are re-assigned to each instance, with higher weights assigned to incorrectly classified instances. The purpose of boosting is to reduce bias and variances. It works on the principle of learners growing sequentially. Each subsequent learner is grown from previously grown learners except the first one. In simple words, weak learners are converted into strong ones [46]. Our model was set with maximum 50 trees.*Decision Tree Classifier* (DC) [47,48]. The DC is a flowchart kind of machine learning algorithm where each node represents a statistical probe on a given attribute. The branches are assigned to the possible outcomes of the probe while the leaf nodes represent a given class label. The paths from the roots to the leaves represent given classification rules. The goal of the DC is to learn rules by making predictions from features. Each branching point in the tree is a rule after which we obtain either the decision itself or a starting point of another subtree. The tree depth defines how many rules need to be applied step by step to obtain the result. In this given case, we did not limit the tree depth during training and all decisions were allowed to use any one of the features.*Gaussian Naive Bayes* (Gaussian) [49] is a very simple machine learning algorithm (also referred to as probabilistic classifier which is based on Bayes’ theorem). In general, the naive Bayes classifiers are highly scalable models, requiring a number of parameters linear in the number of variables in a given learning problem. When dealing with continuous or close-to-continuous data, a typical assumption is that the continuous values are associated with each class that are distributed according to a normal distribution.*K-Nearest Neighbors Classifier* (KNN) [50,51]. This method relies on labels and makes an approximation function for new data. The KNN algorithm assumes that the similar features “fall” close to each other (in numerical sense). The data are represented in a hyperspace based on the characteristics of the data. When new data arrive, the algorithm looks at the *k*-nearest neighbors at the decision. The prediction result is the class that received the most votes based on its neighbors. We use the k−d tree implementation [52] of KNN with k=5.*Support Vector Machines* [53]. This machine learning algorithm can be used for classification, regression, and also for outlier detection (anomaly detection). It is very efficient in high-dimensional spaces and also good when the number of dimensions (features) is greater than the sample number. It is very versatile in the case of using kernels. Common kernels are provided, but we can specify own kernels if we want. The main goal of the algorithm is to find a hyperplane in *N*-dimensional feature space. To separate classes, we can find many different hyperplanes, but the SVM finds the hyperplane which provides the maximum margin. We examined SVMs with 1000 iterations in five kernel variants: radial basis function kernel (SVM kernel = rbf), sigmoid kernel (SVM kernel = sigmoid), 3rd-degree polynomial kernel (SVM kernel = poly degree = 3), 5th-degree polynomial kernel (SVM kernel = poly degree = 5), and 10th-degree polynomial kernel (SVM kernel = poly degree = 10).*Random Forest* [54,55]. The Random Forest is based on decision trees: it creates different trees via training on different random sets of feature vectors from the training set that was selected according to the same rule. In prediction, each tree gives a rating, a vote, which is aggregated to provide a final decision according to the majority of the votes. These votes are aggregated. In our test, the Random Forest was built with 100 trees.*Multi-Layer Perceptron Networks* (MLP) [56,57]. A neuron consists of an input part, which is a vector, the weights of the neuron, which is also a vector, an activation function through which we pass the product of the input vector, and the transposition of the weighting vector. The last element of the neuron is the output, which is the value of the activation function. Activation function can be sigmoid, tangent hyperbolic, Gaussian function, etc. An MLP is realized when multiple neurons are organized next to each other—that is the so-called layer—and multiple layers of neurons are arranged in a row that aggregate the input in a complex way to realize the output of the MLP. The output is obtained by going through the network, layer by layer. The activation function is calculated for each neuron in each layer. In the end, the neuron with the highest value will be the output for that input, i.e., which neuron showed the highest activation for that input.All MLP models involved in this work used four hidden layers of sizes 100, 150, 100, and 50, respectively. They were all trained for maximum 1000 iterations. The deployed MLP models differ in their activation functions, which can be logistic, ReLU, or tanh.

### 2.5. Definition of Use-Cases for Model Development

In the following, we introduce five investigated use-cases, all of which involved tests with the above presented machine learning algorithms.
Recognizing physical activity based only on CGM signal using the cleaned data from the Ohio T1DM dataset (Table 1). Feature vector consists of BG sample points. Data structure: [Date stamp, Time stamp, BG level from CGM (concentration), Exercise (0/1)]. Of the dataset, 75% was used for training, the remaining 25% for testing.Recognizing physical activity based only on CGM signal using the cleaned data from the D1namo dataset. Feature vector consists of BG sample points. Data structure: [Date stamp, Time stamp, BG level from CGM (concentration), Exercise (0/1)]. Of the dataset, 75% was used for training, the remaining 25% for testing.Recognizing physical activity based on CGM and HR signals using the cleaned data from the Ohio T1DM dataset (Table 2). Feature vector consists of BG and HR sample points. Data structure: [Date stamp, Time stamp, BG level from CGM (concentration), HR level (integer), Exercise (0/1)]. Of the dataset, 75% was used for training, the remaining 25% for testing.Recognizing physical activity based on CGM and HR signals using the cleaned data from the D1namo dataset. Feature vector consists of BG and HR sample points. Data structure: [Date stamp, Time stamp, BG level from CGM (concentration), HR level (integer), Exercise (0/1)]. Of the dataset, 75% was used for training, the remaining 25% for testing.Recognizing physical activity based on CGM and HR signals using the cleaned data from the Ohio T1DM (Table 2) and D1namo datasets. Feature vector consists of BG sample points. Data structure: [Date stamp, Time stamp, BG level from CGM (concentration), HR level (integer), Exercise (0/1)]. The train data were the Ohio T1DM data records while the test data were the D1namo data records. All records of the Ohio T1DM dataset were used for training, and all records of D1namo dataset for testing.This use-case can be applied to strengthen our hypothesis as to whether there are many non-reported physical activities in the original Ohio T1DM dataset (where CGM signal was available, physical activity was not reported; however, physical activity possibly happened). Furthermore, this use-case can be applied for data annotation as well on the original Ohio T1DM dataset to select the presumable non-reported physical activities.

### 2.6. Feature Extraction

The feature selection in this study was performed to support the predefined use-cases, namely recognition of physical activity from CGM and from the CGM and HR complex based on different datasets. During the conceptualization, the BG and HR variability dynamics were considered. We made several tests in our previous research [31], and the results show that the usage of the following feature sets are beneficial. We were testing many models based on different algorithms sensitive for different aspects of the data (e.g., differences, velocities, acceleration); thus, we decided to use the features introduced in the following even though the information content in the features are correlating and overlapping.

We defined a 15 records-wide window in which the BG variation was investigated, and the features were extracted—representing a 70 min wide time window. The window was also split into three “mini” windows to analyze the variation in BG levels in the most detailed way.

The *body weight of the patient* (*w*) was applied as “independent” feature. The extracted features from the CGM signal were the following:The *end-to-end difference in the blood glucose level* (*d*) in the window.
(1)d=bg(14)−bg(0)The *blood glucose level difference between two consecutive sampled points* within the window (dp)
(2)dp(i)=bg(i+1)−bg(i),i=0,…,13The *change in the blood glucose levels in the three mini windows*, from beginning to end (dpp)
(3)$$dpp\left(i\right)=\sum_{j=0}^{3}bg\left(5i+j+1\right)-bg\left(5i+j\right)=bg\left(5i+4\right)-bg\left(5i\right),\;\;i=0,\ldots ,2$$*End-to-end blood glucose level change speed* (*v*):
(4)$$v=\frac{bg(14)-bg(0)}{t(14)-t(0)}=\frac{d}{14\cdot \Delta t}$$*The blood glucose level change speed between two consecutive sampled points* (vp)
(5)$$vp\left(i\right)=\frac{bg(i+1)-bg(i)}{t(i+1)-t(i)}=\frac{dp(i)}{\Delta t},\;\;i=0,\ldots ,13$$The *blood glucose level change speed in all mini sliding window measured points*. From beginning to end (vpp)
(6)$$vpp\left(i\right)=\frac{\sum_{j=0}^{3}\left(bg\left(5i+j+1\right)-bg\left(5i+j\right)\right)}{t(5i+4)-t(5i)}=\frac{dpp(i)}{4\cdot \Delta t},\;\;i=0,\ldots ,2$$The *acceleration of blood glucose level change speed among three consecutive sampled points* (ap)
(7)$$ap\left(i\right)=\frac{bg(i+2)-bg(i)}{{\left(t\left(i+2\right)-t\left(i\right)\right)}^{2}},\;\;i=0,\ldots ,12\;.$$

The extracted features from the HR signal were the following:The *HR measured at the CGM sample time* (hr),
(8)hr(i)The *end-to-end heart rate difference*, the difference between the first heart rate measured point and the last heart rate measured point, which point is between two consecutive glucose sample times (hrp),
(9)hrp=hr(i+1)−hr(i).

Due to the selection of the features, especially because of the *d*, the developed models need 15 BG registrations before fully functioning. This gap after initiation of the models is bypassed by using 0 values while the database is “loading”. In this way, the models are functioning; however, precise estimations can be expected after the 15 BG registrations appear, which means 75 min delay from the initiation.

We did not consider the chronological order as a feature; however, we used the data recordings in chronological order which preserves the real-time sequence of the data. The input of the models in this way was coming from the defined features extracted from the cleaned data. When we considered only the CGM recordings, the following feature set was applied: $FS_{1}$ = [*w*, *d*, dp(0), dp(1), dp(2), dp(3), dp(4), dp(5), dp(6), dp(7), dp(8), dp(9), dp(10), dp(11), dp(12), dp(13), dpp(0), dpp(1), dpp(2), *v*, vp(0), vp(1), vp(2), vp(3), vp(4), vp(5), vp(6), vp(7), vp(8), vp(9), vp(10), vp(11), vp(12), vp(13), vpp(0), vpp(1), vpp(2), ap(0), ap(1), ap(2), ap(3), ap(4), ap(5), ap(6), ap(7), ap(8), ap(9), ap(10), ap(11), ap(12)]. When both the CGM and HR recordings were considered, the following feature set was used: $FS_{2}$ = [*w*, *d*, dp(0), dp(1), dp(2), dp(3), dp(4), dp(5), dp(6), dp(7), dp(8), dp(9), dp(10), dp(11), dp(12), dp(13), dpp(0), dpp(1), dpp(2), *v*, vp(0), vp(1), vp(2), vp(3), vp(4), vp(5), vp(6), vp(7), vp(8), vp(9), vp(10), vp(11), vp(12), vp(13), vpp(0), vpp(1), vpp(2), ap(0), ap(1), ap(2), ap(3), ap(4), ap(5), ap(6), ap(7), ap(8), ap(9), ap(10), ap(11), ap(12), hr, hrp]. The models were “fed” with the $FS_{1}$ and $FS_{2}$ sets as inputs during operation.

Figure 2 shows the graphical representation of the way feature extraction introduced above.

### 2.7. Performance Metrics

We have considered the de facto standard evaluation metrics of AI applications [51,58,59]. TP, TN, FP, and FN denote the number of true-positive, true-negative, false-positive, and false-negative results, respectively.

Accuracy (ACC) represents the rate of correct decisions, defined as
(10)$$\mathrm{ACC}=\frac{\mathrm{TP}+\mathrm{TN}}{\mathrm{TP}+\mathrm{TN}+\mathrm{FP}+\mathrm{FN}}\;,$$Recall, also known as sensitivity or true-positive rate (TPR), is defined as
(11)$$\mathrm{TPR}=\frac{\mathrm{TP}}{\mathrm{TP}+\mathrm{FN}}\;,$$Specificity, also known as true-negative rate (TNR), is defined as
(12)$$\mathrm{TNR}=\frac{\mathrm{TN}}{\mathrm{TN}+\mathrm{FP}}\;,$$Precision, also known as positive prediction value (PPV), is defined as
(13)$$\mathrm{PPV}=\frac{\mathrm{TP}}{\mathrm{TP}+\mathrm{FP}}\;,$$False-positive rate (FPR) is defined as
(14)$$\mathrm{FPR}=\frac{\mathrm{FP}}{\mathrm{TN}+\mathrm{FP}}\;,$$$F_{1}$-score ($F_{1}$), also known as Dice score, is defined as
(15)$$F_{1}=\frac{2\cdot \mathrm{TP}}{2\cdot \mathrm{TP}+\mathrm{FP}+\mathrm{FN}}\;.$$

Besides all the above introduced statistical indicators, the AUC metric [60] based on the ROC curve was applied in order to assess the performance of the different classifiers.

## 3. Results

The results are presented in accordance with the predefined five use-cases. The benchmarks presented in the following are based on the test datasets belonging to the different use-cases.

### 3.1. Ohio T1DM Dataset Using Only Blood Glucose Level Features—Use-Case 1

We have investigated the performance of the machine learning methods introduced in Section 2.4 using the extracted features defined for use-case 1. The results—based on the metrics defined—can be seen in Figure 3 and Table 4.

The results in this case did not achieve the accuracy we expected based on the previous findings [31]. In general, a model performance can be acceptable if the AUC is higher than 0.8 [51]; however, by using only the CGM signal from the Ohio dataset, we were able to achieve only 0.65, which was produced by the “best” performing model—the Random Forest in the current examination. The average AUC value of the set of the various models was 0.57.

We can see similar results in the defined metrics in Table 4 as we experienced by analyzing the AUC diagram (Figure 3). The table shows that 0.602 is the highest TPR score reached by the Random Forest and AdaBoost models, indicating that on this dataset using these features, model architectures, and only the CGM signals, we can predict the physical activity with 60% probability. The best PPV value of 0.657 was reached by the Decision Tree.

The results reflect that the models incorrectly predict the positive class and the presence of physical activity several times. The highest *F*_1_ score of 0.632 was achieved by the AdaBoost model. Eventually, the “best” model in this context was the Random Forest with $F_{1}=0.632$, TPR=0.602, and PPV=0.643; however, these results suggest that we have almost a 40% error rate in the prediction.

### 3.2. D1namo Dataset Using Blood Glucose Level Features Only—Use-Case 2

We have investigated the performance of the same models using the D1namo dataset as well in order to exclude the dataset dependency. The obtained ROC curves are exhibited in Figure 4 while the detailed statistical metrics are listed in Table 5.

Figure 4 shows the ROC plots regarding the tested models. The best achieved AUC value is 0.72, which exceeds the AUC score obtained in the Ohio T1DM dataset test using the CGM data only. The average AUC value of all the models is 0.62; thus, using only the CGM data to identify physical activity is possible but with a poor performance according to this metric tested on both datasets. It should be noted that we found it really challenging to extract the right threshold value from classifying the dataset.

The results according to the different metrics are presented in Table 5. Some of the models provided TPR=1, which means they classified all the positive data correctly, but they were unable to classify the negative ones correctly. The models that have $F_{1}>0.6$ can be considered acceptable ones according to the field of usage in order to continue their refinement. The models achieving $F_{1}<0.5$ can be considered unsatisfactory. The KNN performed best according to the *F*_1_ score (0.77), seconded by the SVM model with the 5th-order polynomial kernel at a slight difference (0.758), and followed by poor *F*_1_ scores achieved by all the other models.

The combination of TPR=0.667 and TNR=0.910 achieved by the KNN model means that it misclassifies one-third of the positive cases and only 9% of the negative ones. On the other hand, the best performing SVM model detects all the positive cases (TPR=1) but misses almost 40% of the negative ones (TNR=0.611), which represents a poor result. The relatively high TNR and PPV values make the KNN model the best one in this use-case.

### 3.3. Ohio T1DM Dataset with Blood Glucose and Heart Rate Features—Use-Case 3

The results are presented in the same way as in the previous sections; they can be seen in Figure 5 and Table 6.

Figure 5 shows the AUC performance of the assessed models using the Ohio T1DM dataset with features extracted from the CGM and HR signals. Our hypothesis was that the latest HR data reflect fast changes in physical activity, regardless of the nervous activity, simply due to the stress caused by the muscle usage, and comparing this signal with the CGM data, a better performance can be achieved. The other thought behind the selection of the HR was that these sensors with accessible data are quite frequent and available to the diabetic community. In the presence of the CAN, the deep usage of the HR can lead to false predictions; thus, we applied limited information from the HR signal. By using this additional information, the performance of the models improved significantly.

The highest AUC results exceeded 0.9; they were produced by the AdaBoost (0.91) and the Random Forest (0.90) models. Compared to the previous AUC results based on the CGM signals only, the average AUC improved by almost 0.3. The LR model produced a surprisingly good AUC compared to the previous 0.59.

Table 6 shows the results based on the predefined metrics. Comparing the achievements of the models based only on the CGM signal, there is a significant improvement in all the columns (see Table 4). Similar to the AUC values, the best TPR result of 0.850 belongs to the LR model. Thus, the LR model in this context reaches 85% of “good” classification (namely 15% misclassification). It should be noted that the joint usage of the CGM and HR signals provided similar results to what we experienced in our previous work using only the CGM, however, in a synthetic environment [31].

### 3.4. D1namo Dataset with Blood Glucose and Heart Rate Features—Use-Case 4

The results of the tests made in conjunction with use-case 4 can be seen in Figure 6 and Table 7.

Figure 6 represents the ROC curves and corresponding AUC values for the tested models, when we used both the features extracted from the CGM and HR data of the D1namo dataset. We experienced the same performance enhancement as in the previous case, using both the CGM and HR originated features. According to Table 7, the Random Forest provided the best results here; however, the LR reached an almost identical performance.

It should be noted that the D1namo dataset has less records than the Ohio T1DM dataset; however, the experiences are quite the same.

In this section, we provided the results when both the CGM and HR features were applied from both datasets; however, the training dataset was the Ohio T1DM and the test dataset was the D1namo. Namely, the models were trained only on the Ohio T1DM dataset while the tests were executed on the D1namo dataset. Thus, this use-case realizes a robust test of the hypothesis behind the research, and depending on the results, two questions can be answered: are the models robust enough based on the defined metrics and, furthermore, are the models applicable to use them to annotate the Ohio T1DM dataset with the given performance?

Figure 7 shows the results of the tests. It can be seen that the LR and AdaBoost methods provided the best AUC (exceeding 0.9), and further on, the Random Forest and two MLP models also reached an acceptable AUC-related performance.

### 3.5. Experiences with Use-Case 5

Table 8 shows the performance of the models. Five models reached an AUC≥0.8, namely the Logistic Regression, AdaBoost, Random Forest, MLP with ReLU activation function, and MLP with Tanh activation function; thus, the evaluation focuses on these models in the following. The Logistic Regression achieved an ACC=0.845, which is the highest value among the models. The most important metric, the TPR, shows a 0.818 value here; thus, the model can predict almost 82% of positive cases. The PPV result is 0.864, indicating that the LR model provides a mistaken decision in only 14% of the cases when it predicts positive. The PPV value of the LR is also the highest among all the models.

The *F*_1_ score of the LR model turned out to be the best with 0.844, which is less than 0.9 as achieved in the previous use-case.

The AdaBoost model reached 0.909 of the TPR score which is better than all the other models in the test. Thus, the AdaBoost is able to detect well the positive classes but made some mistakes while predicting the negative classes. The PPV value is 0.80, which is 6% below the precision of the LR. The *F*_1_ score of this model was 0.838 which is close to the performance of the LR.

The Random Forest model achieved 0.818 in all of the following metrics: the TPR, ACC, PPV, and *F*_1_ score, which is an unexpected coincidence. That means the model is capable of correctly detecting 82% of the physical activity by making 18% mistakes when predicting the negative class.

Surprisingly, the MLP variants did not give as fine predictions as the other algorithms discussed above, with neither of the tested activation functions.

The results show that the AdaBoost algorithm is the “best” in this test case.

## 4. Discussion

We have investigated different use-cases in this study in order to establish different, simple AI models to accurately detect physical activity.

The original idea was to use only the CGM signal, because the CGM sensor can be found in the daily routine of many T1DM patients. According to our previous work using simulated patient data [31], it is possible to detect physical activity from the CGM with good performance metrics (by definition of what a “good” AI model means [51]).

We have to remark that the case here is similar to any other medical applications in that the sensitivity (recall) of the models should be as good as possible. Namely, if the solutions are applied, for example, to make automated corrections in the insulin dosage regimen of an insulin pump or to suggest a correction during manual therapy, then this metric is always the most important. In this particular case, this means that in the absence of physical activity, however, the algorithm may mistakenly “detect” physical activity and suggest lowering the insulin dosage and thus possibly cause hyperglycemia, which is a bearable condition where intervention according to the BG trend is always possible. Nevertheless, if there is physical activity, but the algorithm does not recognize it, it can happen that the regular dosage of the administered insulin leads to hypoglycemia, which is a dangerous condition that can be tolerated only in the short term. Due to the dynamics of the metabolic system, the processes are slow enough to intervene before a highly dangerous condition appears.

However, the solutions shall be established to reach as high a sensitivity as possible. We performed our analysis in the spirit of the aforementioned conditions.

This study was built upon the conclusions of our previous work analyzing the developments and tested models [31]; however, previously, we applied synthetic data (based on tests relying on virtual patient models) only.

According to our investigations in Section 3.1 and Section 3.2, we can conclude that the detection of physical activity based on the features originating from the CGM is only possible as we expected, in accordance with our previous research based on synthetic data (a virtual patient simulation) [31]. However, the AUC and other statistical indicators showed that the prediction capabilities are poorer compared to the goals of the research, and the involvement of further physiological signals should be investigated.

The further results of Section 3.3 and Section 3.4 showed that our hypothesis was good, and with extra HR features, we can obtain an equal performance to what we have achieved in the previous work when we have used only the glucose data and a virtual patient [31].

Furthermore, the last tests we made related to use-case 5 showed that the developed models are robust, and it is possible to reach an acceptable performance by training the models on the Ohio T1DM dataset while making performance tests on the D1namo dataset (two datasets with different patients and different sensors).

The AUC-based summary of all the tests regarding all use-cases 1–5 can be seen in Figure 8 and Table 9. According to the findings in Section 3, the LR, AdaBoost, and Random Forest turned out to be the most promising algorithms—these models will be further developed in our future work.

## 5. Conclusions

We can conclude that physical activity detection using machine learning algorithms, relying on real CGM and CGM completed with HR data as inputs, is possible based on the predefined circumstances. We also proved that the usage of additional HR-based features raised the best achievable AUC values from 0.65 to 0.91 in the case of the Ohio T1DM dataset and from 0.72 to 0.92 in the case of the D1namo dataset, which represents a significant improvement in the results. It should be noted that the measurement of the HR is quite regular and can be solved by using low-cost wearable devices. However, the HR can be affected by the CAN appearing as a result of the diabetic condition. This fact should be considered during the usage of the proposed developments.

We have shown that when using the blood glucose data of real patients, we need additional features, such as the ones extracted from heart rate signals. The Logistic Regression, AdaBoost, Random Forest, and Multi-Layer Perceptron with the ReLU and Tanh activation function were found to be the best five models from our tests; they provide better or comparable results to those reported in similar studies, e.g., [61,62,63,64]. The other models require a post-processing of the predicted data to be more precise. The least accurate models produce too many false positives.

We also made a test using mixed real patient datasets: Ohio T1DM was the training dataset and D1namo was the testing dataset. This test is a significant point of our research, because we achieved a good performance on different populations, different clinical trials for patients, and different CGM and heart rate sensors. So, from this test, we can conclude that the models which were the best in the test cases will be useful for other populations with other sensors as well; the only things needed are the CGM sensor and heart rate sensor. In the future, it will be of interest to investigate what happens if we use more sensors.

In our future work, we will test further models that are capable of considering the temporary nature of the data, e.g., deep long short-term memory (LSTM) models and deep leaning models with multiple input features, namely the input separation will be performed before applying the models.

## Figures and Tables

**Figure 1 sensors-22-08568-f001:**
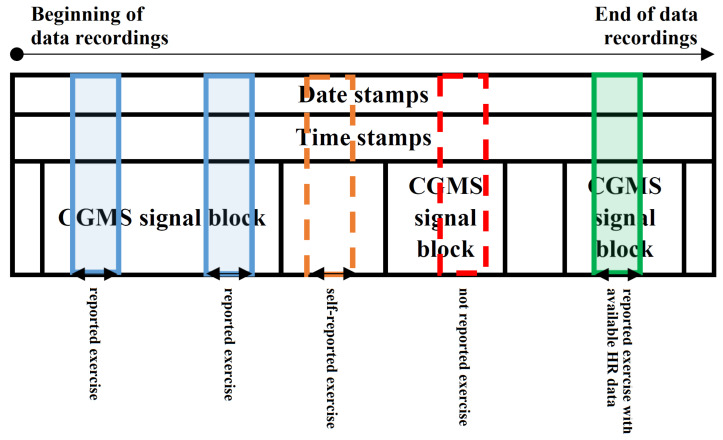
The ways of data extraction from the Ohio T1DM dataset for a given patient. Black arrows indicate 24 h long blocks according to the time stamps, midnight to midnight. Legend: blue blocks—CGM data available, exercise reported, no HR data available; green blocks—CGM data available, exercise reported, HR data available; transparent orange area—self-reported exercise, no CGM data available; transparent red area—probably an exercise event happened, but it was not reported.

**Figure 2 sensors-22-08568-f002:**
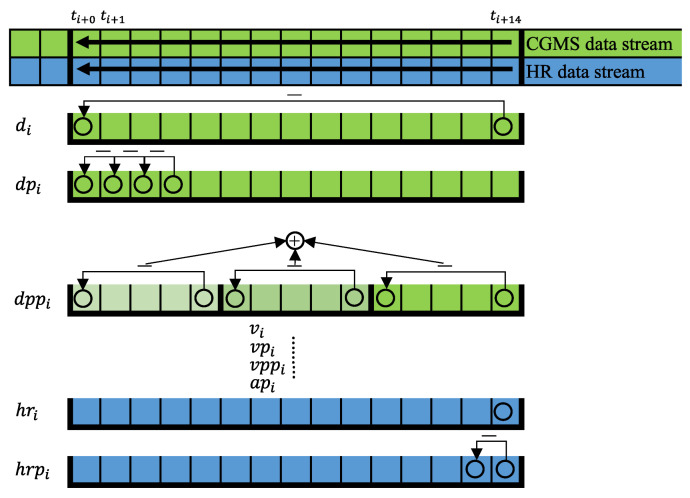
Abstraction of feature extraction during operation. The *v*, vp, vpp, and ap features originate from the *d*-kind features; thus, these are not listed here. In the case of the $dp_{i}$, only the first four values are represented as a demonstration; however, all sampled values are considered from the window during operation.

**Figure 3 sensors-22-08568-f003:**
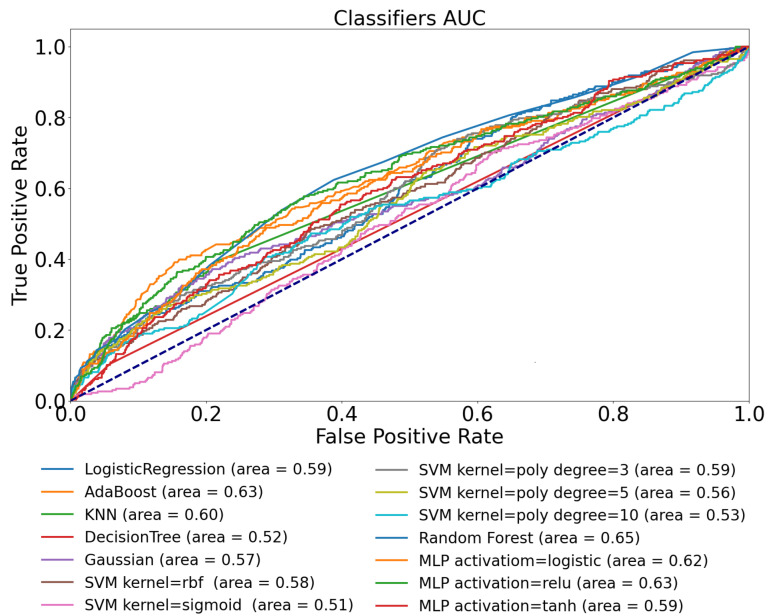
ROC curve of tested ML models for Ohio T1DM dataset using glucose features only.

**Figure 4 sensors-22-08568-f004:**
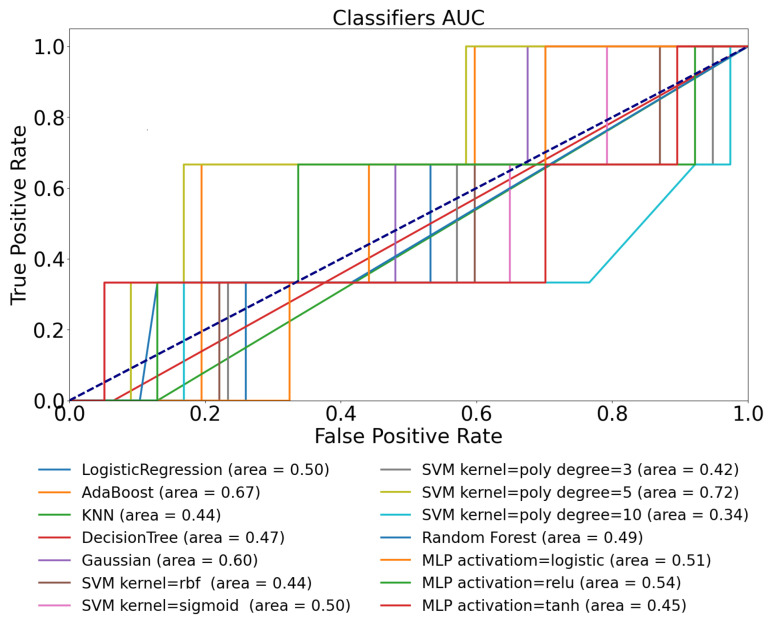
ROC curves of various ML models obtained on D1namo dataset using glucose features only.

**Figure 5 sensors-22-08568-f005:**
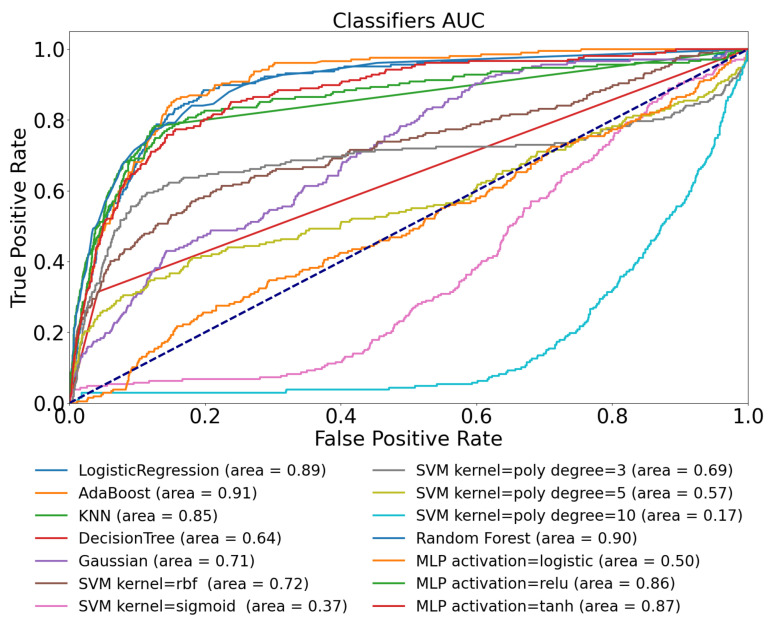
ROC curves of models on Ohio T1DM dataset using both blood glucose and heart rate features.

**Figure 6 sensors-22-08568-f006:**
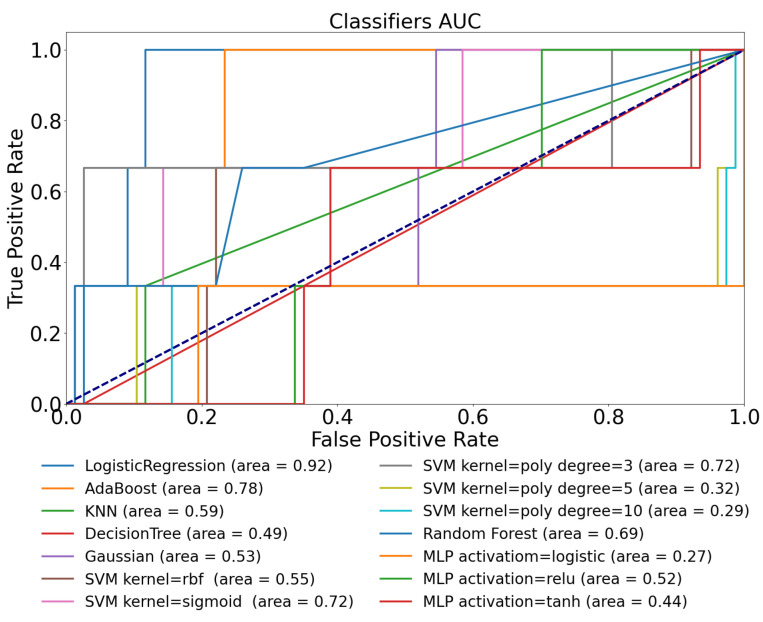
ROC curves of models obtained on D1namo dataset with blood glucose and heart rate features.

**Figure 7 sensors-22-08568-f007:**
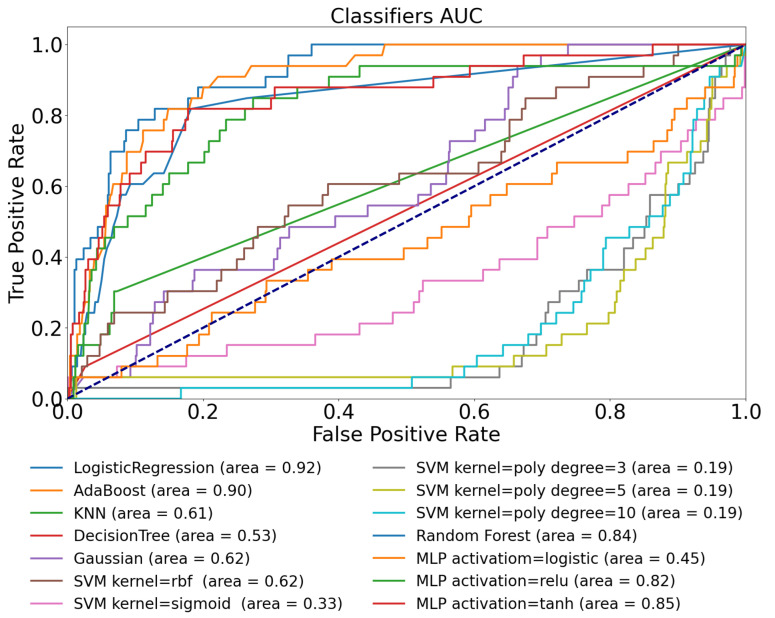
ROC curve of models on Ohio T1DM is the training data; the D1namo is the test with glucose and heart rate feature.

**Figure 8 sensors-22-08568-f008:**
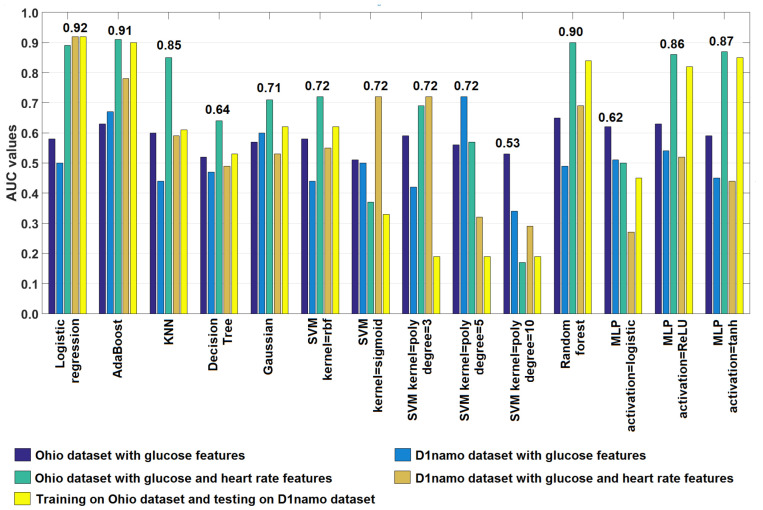
AUC of models in all use-cases—test results.

**Table 1 sensors-22-08568-t001:** Properties of the cleaned data from the Ohio T1DM dataset used for patients’ blood glucose feature extraction.

Ohio T1DM Patient (OP) ID	Number of Records	Total Time Duration in Days
OP3	546	2
OP8	2730	10
OP9	546	2
OP10	3276	12
OP14	2730	10
OP18	546	2
OP19	7917	29
OP20	819	3
OP21	1638	6
OP22	1365	5
OP23	4914	18
Total	27,027	99

**Table 2 sensors-22-08568-t002:** Properties of the cleaned data from the Ohio T1DM dataset used for the extraction of patients’ blood glucose and heart rate features.

Ohio T1DM Patient (OP) ID	Number of Records	Total Time Duration in Days
OP8	2730	10
OP9	546	2
OP10	3276	12
OP14	2730	10
OP18	546	2
OP19	7917	29
OP20	819	3
OP21	1638	6
Total	20,202	74

**Table 3 sensors-22-08568-t003:** Properties of the cleaned data from the D1namo dataset used for the extraction of patients’ blood glucose and heart rate features.

D1namo Patient	Record Count for CGM Features	Record Count for CGM and HR Features	Total Time Duration in Days
Patient 1	939	858	6
Patient 2	1575	1440	5
Patient 3	185	166	2
Patient 4	1383	1266	4
Patient 5	1375	1251	4
Patient 6	1225	1108	6
Patient 7	966	879	5
Patient 8	1189	1088	5
Patient 9	135	85	2
Total	8972	8141	39

**Table 4 sensors-22-08568-t004:** Performance score on Ohio T1DM dataset using blood glucose feature only.

Classifier	ACC	TPR	TNR	PPV	*F* _1_
Logistic Regression	0.566	0.487	0.645	0.578	0.555
AdaBoost	0.634	0.602	0.666	0.643	0.632
KNN	0.598	0.411	0.784	0.656	0.539
Decision Tree	0.527	0.114	0.940	0.657	0.204
Gaussian	0.602	0.496	0.708	0.629	0.583
SVM kernel = rbf	0.514	0.462	0.566	0.515	0.509
SVM kernel = sigmoid	0.506	0.500	0.512	0.506	0.506
SVM kernel = poly, deg = 3	0.536	0.581	0.492	0.533	0.532
SVM kernel = poly, deg = 5	0.509	0.547	0.471	0.508	0.506
SVM kernel = poly, deg = 10	0.487	0.377	0.597	0.483	0.462
Random Forest	0.611	0.602	0.619	0.612	0.610
MLP activation = logistic	0.609	0.589	0.629	0.614	0.608
MLP activation = ReLU	0.610	0.547	0.674	0.626	0.604
MLP activation = tanh	0.579	0.593	0.566	0.577	0.579

**Table 5 sensors-22-08568-t005:** Performance on D1namo dataset using blood glucose features only.

Classifier	ACC	TPR	TNR	PPV	*F* _1_
Logistic Regression	0.606	0.667	0.546	0.595	0.600
AdaBoost	0.727	1.000	0.454	0.647	0.625
KNN	0.788	0.667	0.910	0.881	0.770
Decision Tree	0.494	0.000	0.988	0.000	0.000
Gaussian	0.669	1.000	0.337	0.602	0.504
SVM kernel = rbf	0.669	1.000	0.337	0.602	0.504
SVM kernel = sigmoid	0.403	0.000	0.806	0.000	0.000
SVM kernel = poly, deg = 3	0.537	0.333	0.741	0.563	0.460
SVM kernel = poly, deg = 5	0.805	1.000	0.611	0.720	0.758
SVM kernel = poly, deg = 10	0.602	0.333	0.871	0.721	0.482
Random Forest	0.708	1.000	0.415	0.631	0.587
MLP activation = logistic	0.606	0.667	0.546	0.595	0.600
MLP activation = ReLU	0.697	0.667	0.728	0.710	0.696
MLP activation = tanh	0.695	1.000	0.389	0.621	0.561

**Table 6 sensors-22-08568-t006:** Performance on Ohio T1DM dataset with glucose and heart rate feature.

Classifier	ACC	TPR	TNR	PPV	*F* _1_
Logistic Regression	0.832	0.850	0.814	0.820	0.831
AdaBoost	0.827	0.819	0.835	0.832	0.827
KNN	0.821	0.768	0.874	0.859	0.817
Decision Tree	0.615	0.275	0.954	0.857	0.427
Gaussian	0.632	0.544	0.720	0.660	0.619
SVM kernel = rbf	0.666	0.712	0.621	0.652	0.663
SVM kernel = sigmoid	0.621	0.664	0.578	0.611	0.618
SVM kernel = poly, deg = 3	0.466	0.465	0.466	0.466	0.466
SVM kernel = poly, deg = 5	0.287	0.435	0.138	0.336	0.210
SVM kernel = poly, deg = 10	0.328	0.409	0.247	0.352	0.308
Random Forest	0.827	0.811	0.843	0.838	0.827
MLP activation = logistic	0.536	0.443	0.629	0.544	0.520
MLP activation = ReLU	0.779	0.768	0.791	0.786	0.779
MLP activation = tanh	0.807	0.789	0.825	0.818	0.806

**Table 7 sensors-22-08568-t007:** Performance D1namo dataset with glucose and heart rate features.

Classifier	ACC	TPR	TNR	PPV	*F* _1_
Logistic Regression	0.922	1.000	0.845	0.866	0.916
AdaBoost	0.792	1.000	0.585	0.707	0.738
KNN	0.801	0.667	0.936	0.912	0.779
Decision Tree	0.654	0.333	0.975	0.930	0.497
Gaussian	0.439	0.333	0.546	0.423	0.414
SVM kernel = rbf	0.511	0.333	0.689	0.517	0.449
SVM kernel = sigmoid	0.736	0.667	0.806	0.774	0.730
SVM kernel = poly, deg = 3	0.814	0.667	0.962	0.946	0.788
SVM kernel = poly, deg = 5	0.717	0.667	0.767	0.741	0.713
SVM kernel = poly, deg = 10	0.743	0.667	0.819	0.786	0.735
Random Forest	0.945	1.000	0.889	0.901	0.941
MLP activation = logistic	0.593	0.667	0.520	0.581	0.584
MLP activation = ReLU	0.589	0.333	0.845	0.682	0.478
MLP activation = tanh	0.762	0.667	0.858	0.824	0.750

**Table 8 sensors-22-08568-t008:** Performance of various classifier algorithms in the case when trained on Ohio T1DM dataset and tested on D1namo using both blood glucose and heart rate features.

Classifier	ACC	TPR	TNR	PPV	*F* _1_
Logistic Regression	0.845	0.818	0.871	0.864	0.844
AdaBoost	0.844	0.909	0.778	0.804	0.838
KNN	0.617	0.303	0.931	0.814	0.457
Decision Tree	0.522	0.061	0.983	0.778	0.114
Gaussian	0.579	0.485	0.673	0.597	0.564
SVM kernel = rbf	0.611	0.606	0.616	0.612	0.611
SVM kernel = sigmoid	0.649	0.606	0.693	0.664	0.647
SVM kernel = poly, deg = 3	0.299	0.364	0.234	0.322	0.285
SVM kernel = poly, deg = 5	0.391	0.667	0.114	0.429	0.195
SVM kernel = poly, deg = 10	0.330	0.455	0.206	0.364	0.284
Random Forest	0.818	0.818	0.818	0.818	0.818
MLP activation = logistic	0.502	0.394	0.610	0.503	0.479
MLP activation = ReLU	0.787	0.848	0.726	0.756	0.783
MLP activation = tanh	0.781	0.697	0.866	0.838	0.772

**Table 9 sensors-22-08568-t009:** AUC performance achieved in different tests.

Training Dataset	Ohio T1DM	D1namo	Ohio T1DM	D1namo	Ohio T1DM
**Testing Dataset**	**Ohio T1DM**	**D1namo**	**Ohio T1DM**	**D1namo**	**D1namo**
**Features**	**BG Only**	**BG Only**	**BG and HR**	**BG and HR**	**BG and HR**
Logistic Regression	0.58	0.50	0.89	0.92	0.92
AdaBoost	0.63	0.67	0.91	0.78	0.90
KNN	0.60	0.44	0.85	0.59	0.61
Decision Tree	0.52	0.47	0.64	0.49	0.53
Gaussian	0.57	0.60	0.71	0.53	0.62
SVM kernel = rbf	0.58	0.44	0.72	0.55	0.62
SVM kernel = sigmoid	0.51	0.50	0.37	0.72	0.33
SVM kernel = poly, degree = 3	0.59	0.42	0.69	0.72	0.19
SVM kernel = poly, degree = 5	0.56	0.72	0.57	0.32	0.19
SVM kernel = poly, degree = 10	0.53	0.34	0.17	0.29	0.19
Random Forest	0.65	0.49	0.90	0.69	0.84
MLP activation = logistic	0.62	0.51	0.50	0.27	0.45
MLP activation = ReLU	0.63	0.54	0.86	0.52	0.82
MLP activation = tanh	0.59	0.45	0.87	0.44	0.85

## Data Availability

Not applicable.

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
