# Peer review of "Detection of Physical Activity Using Machine Learning Methods Based on Continuous Blood Glucose Monitoring and Heart Rate Signals"

_sensors, 2022, doi:10.3390/s22218568_

Round 1

Reviewer 1 Report

In this work, the authors presented the detection of physical activity using a continuous glucose monitoring system and heart rate. The results showed that the detection of physical activity was possible based on these data sources using machine learning methods. The reached approximately 90% of performance in the detection of physical activity with the models. They found various machine learning algorithms that performed well (e.g. KNN, 118 Random Forest, Decision Tree) in the physical activity detection. Overall, the content of this work is somewhat interesting. The suggestions are as follows.

(1) The abstract section has too much background description. The authors should provide some key useful review information to capture the readers' eyes.

(2) The background introduction is too much. The authors should briefly and clearly introduce the development and existing problems of the relevant research.

(3) Please indicate the full name of abbreviations when they were first used in the manuscript.

(4) Many curved signals are cluttered, and even the content is not clear, such as Figures 3, 5, and 7. Authors should improve them.

(5) The working mechanism should be drawn, so that the readers would know them better.

(6) What are the different between this work and others, such as methods, the advantages and disadvantages of performance? The authors should compare them with necessary description to explain the innovation of this work.

Author Response

First of all, on behalf of the authors, I would like to thank the work of the Reviewer and the thorough comments.

We changed the manuscript as it was asked.

Please find the exact answers below.

Best regards,

György Eigner

(1) The abstract section has too much background description. The authors should provide some key useful review information to capture the readers' eyes.

The abstract was completely rewritten. Now it is shorter and more focused.

(2) The background introduction is too much. The authors should briefly and clearly introduce the development and existing problems of the relevant research.

We did our best to shorten the presentation of background work.

(3) Please indicate the full name of abbreviations when they were first used in the manuscript.

We did our best to introduce all abbreviations at first use and to include them into the table of abbreviations.

(4) Many curved signals are cluttered, and even the content is not clear, such as Figures 3, 5,and 7. Authors should improve them.

Thank you for emphasizing this. Figures 1, 3-8 were redrawn to make the contents more accessible.

(5) The working mechanism should be drawn, so that the readers would know them better.

Thank you for your suggestion. In a nutshell, the operation paradigm is the following. The CGMS an HR signals are gathered and stored in one common database. The resolution of the CGMS signal is 5 minutes -- thus the system gets BG data every 5 minutes. The resolution of the HR data is 5 times faster (HR data in every 1 minute). The HR data is compressed and aggregated (using the mathematical mean) to be available in every 5 minutes. In this way, the resolution of the data is the same. We use 15 data points from the CGMS, thus the model needs 15 BG values for fully functioning. That means the operation of the model starts with a loading period that takes approx. 75 minutes (remember, we do have a BG value every 5 minutes, thus we need 15x5 minutes to have 15 BG samples). We use only 2 HR values (takes 10 minutes to have them), the model considers only the last two HR values, since the HR signal is much faster than the variation of the BG level. We transform the CGMS signal to velocity and acceleration of the BG level changing as well. We apply these inputs in the model. After the loading period, the model is fully functional, namely, it is capable to predict whether a physical activity happens in each iteration (namely, at each time stamp) after 75 minutes.

(6) What are the differences between this work and others, such as methods, and the advantages, and disadvantages of performance? The authors should compare them with necessary description to explain the innovation of this work.

This is an important comment, thank you for the reviewer. We made a short literature review in this regard with available solutions at the beginning of the paper. Besides, the main advantage of the paper is not in the applied methods, instead, the data processing paradigm and the usage of the data to feed the model. We consider the HR values as well, not only the CGMS. However, most of the solutions need more data than our concept or do not take into account physical activities. Our solution is usually more precise than others in the literature [61-64].

Reviewer 2 Report

A well presented paper. some suggestions:

a) I understand that this paper only selected CGMS + HR as features. It would still be good to discuss its performance agains those using IMU units. The reason is that I feel the accuracy of using only CGMS+HR is not high on some cases;

b) It may not necessary to list details of various ML methods, especially displaying their formula. Because then you need to explain meanings of every parameter. All these information can be easily found at any ML textbooks.

c) CGMS, please explain it the first time it appears on Page 2.

d) Is this a real-time monitoring system? If so, maybe some experiments on algorithms' ability to detect activities on the fly.

Author Response

First of all, on behalf of the authors, I would like to thank the work of the Reviewer and the thorough comments.

We changed the manuscript as it was asked.

Please find the exact answers below.

Best regards,

György Eigner

a) I understand that this paper only selected CGMS + HR as features. It would still be good to discuss its performance against those using IMU units. The reason is that I feel the accuracy of using only CGMS+HR is not high on some cases.

 Thank you for the comment - IMU data can be considered. In this study, our aim was to use as less sensor input as possible from devices that are usually available or cheap enough from the user's point of view. IMU data are hardly accessible, and the preprocessed data is not always trustable due to the applied filtering techniques are not public from the manufacturer's side. The HR data is accessible and usually, the calculation paradigm is known due to open standards (making the connection between the emission-transmission signals and the HR parameter). HR is a good and fast indicator of doing sports activities and stress as well. On the other hand, the CGMS signal and the HR signal are not far from each other from a reasonable resolution point of view with respect to physiology. That means, the CGMS signal is available with a 5 minutes resolution in almost all devices and the bigger change in the HR can be seen on the second's scale. Thus, the usage of the two signals is reasonable here.

 b) It may not necessary to list details of various ML methods, especially displaying their formula. Because then you need to explain meanings of every parameter. All these information can be easily found at any ML textbooks.

Thank you for the valuable comment. Some details including the formulas were removed from the revised version.

 c) CGMS, please explain it the first time it appears on Page 2.

 The abbreviation is introduced at first use in the revised version.

 d) Is this a real-time monitoring system? If so, maybe some experiments on algorithms' ability to detect activities on the fly.

 The system was developed in such a way that it can be used for active monitoring. However, this investigation will be in our future work. Óbuda University in a consortium with two Hungarian hospitals got the permission of the Hungarian Ethical Committee to make an investigation the correlation between CGMS and HR, making the modeling part more solid. We will introduce the results of these trials in our upcoming work.

Reviewer 3 Report

- The figure caption for Figure 1 is very long. I think the sentences after "The ways of data extraction from the Ohio T1DM Dataset at a given patient – graphical example." should be written as a new paragraph.

- Figure 1 seems complicated and hard to understand. Figure 1 should be modified. And why is the first "CGMS signal block" wider than the other ones?

- There are some typo errors (e.g. "the mot suitable" Line 239, "the algorithm to is find" Line 278). The article needs to be checked in this context.

- Equation 4 contains parenthesis errors. The equation should be checked again.

- The information in brackets in lines 322 and 323 is not needed and can be deleted.

- The figure caption for Figure 2 is very long. I think the sentences after the "Abstraction of feature extraction during operation." should be written as a new paragraph.

- Figure 3 and the sentence in line 418 should be moved to Results.

- The font of the text used in the figures should be the same as the font of the article.

- The Y-axis title of Figure 8 is missing.

- The conclusion section should be supported with numerical results.

Author Response

First of all, on behalf of the authors, I would like to thank the work of the Reviewer and the thorough comments.

We changed the manuscript as it was asked.

Please find the exact answers below.

Best regards,

György Eigner

- The figure caption for Figure 1 is very long. I think the sentences after "The ways of data extraction from the Ohio T1DM Dataset at a given patient – graphical example." should be written as a new paragraph.

The caption of Figure 1 was reduced, and it still says everything that is necessary.

- Figure 1 seems complicated and hard to understand. Figure 1 should be modified. And why is the first "CGMS signal block" wider than the other ones?

Since the figure is an illustration, thus is no specific reason why the first block of CGMS data is longer than the later ones. The figure depicts the abstraction of data and helps to understand the main difficulties as well, namely, we have CGMS data but not at all times due to sensor issues, lack of available sensor,s or data failure. 

- There are some typo errors (e.g. "the mot suitable" Line 239, "the algorithm to is find" Line278). The article needs to be checked in this context

We made the correction.

- Equation 4 contains parenthesis errors. The equation should be checked again.

We made the correction. The equation was removed from the revised version.

- The information in brackets in lines 322 and 323 is not needed and can be deleted.

We made the correction.

- The figure caption for Figure 2 is very long. I think the sentences after the "Abstraction of feature extraction during operation." should be written as a new paragraph.

Thank you for the comment. The caption is needed for understanding the figure in our opinion. With all due respect, we did not change it, since many details of the figure can be interpreted only with the caption.

- Figure 3 and the sentence in line 418 should be moved to Results.

We made the correction.

- The font of the text used in the figures should be the same as the font of the article.

We did our best to use readable fonts everywhere.

- The Y-axis title of Figure 8 is missing.

Figure 8 was completely redrawn, not only to comply with the valuable comment.

- The conclusion section should be supported with numerical results.

Best achieved AUC values were included to support the conclusions. 

Round 2

Reviewer 1 Report

There is no other suggestion.